

# Smoking close to others and butt littering at bus stops: pilot observational study

Nick Wilson, Jane Oliver and George Thomson

Department of Public Health, University of Otago, Wellington, Wellington, New Zealand

## ABSTRACT

**Background.** Transportation settings such as bus stops and train station platforms are increasingly the target for new smokefree legislation. Relevant issues include secondhand smoke exposure, nuisance, litter, fire risks and the normalization of smoking. We therefore aimed to pilot study aspects of smoking behavior and butt disposal at bus stops.

**Methods.** Systematic observation of smoking and butt disposal by smokers at bus stops. The selection of 11 sites was a mix of convenience and purposeful (bus stops on main routes) in two New Zealand cities.

**Results.** During 27 h of observation, a total of 112 lit cigarettes were observed being smoked. Smoking occurred in the presence of: just adults (46%), both young people and adults (44%), just young people (6%) and alone (5%). An average of 6.3 adults and 3.8 young people were present at the bus stops while smoking occurred, at average minimum distances of 1.7 and 2.2 m respectively. In bus stops that included an enclosed shelter, 33% of the cigarettes were smoked inside the shelter with others present. Littering was the major form of cigarette disposal with 84% of cigarettes smoked being littered (95% CI; 77%–90%). Also, 4% of disposals were into vegetation, which may pose a fire risk.

**Conclusions.** This pilot study is limited by its small size and various methodological aspects but it appears to be a first attempt to provide observational evidence around smoking at bus stops. The issues described could be considered by policy makers who are investigating national smokefree laws or by-laws covering transportation settings.

Corresponding author
Nick Wilson, nick.wilson@otago.ac.nz

## INTRODUCTION

Secondhand smoke exposure is an important part of the risk posed by tobacco use globally (*Lim et al., 2012*). Part of this problem arises from smoking in transportation settings such as bus stops and train station platforms (*Platt et al., 2009*; *Tan, 2013*). In New Zealand, national survey data indicates that secondhand smoke (SHS) exposure at a bus stop or train station is also sometimes reported (11.9% of respondents reporting exposure in the last month; 95% CI; 10.6–13.2%) (*Wilson, Edwards & Parry, 2011*). Other survey data indicates that public transport users have nuisance and health concerns around being exposed to SHS in these settings (*Russell, Wilson & Thomson, 2012*).

Furthermore, some markedly elevated levels of fine particulates from tobacco smoke have been found in New Zealand bus shelters that were enclosed and partially enclosed (*Patel, Thomson & Wilson, 2012*). The only relevant smoking prevalence data that we found indicated an 11% point prevalence in New Zealand outdoor transport waiting areas (and 7% in England and Scotland). This was for airport entrances, train stations, and bus stations, as well as street bus stops (*Thomson et al., 2013*).

Another problem is from smoking-related litter at bus stops which contributes to cleaning costs, fire risks and may make the experience of using public transport less attractive. There is some evidence that a majority of smokers litter their cigarette butts (*Rath et al., 2012*; *Patel, Thomson & Wilson, 2013*), but the extent of this in the bus stop setting does not appear to have been documented.

In response to these issues, transportation settings such as bus stops and train station platforms are increasingly being covered by smokefree legislation in various jurisdictions (*Tasmanian Parliament, 2012*; *American Nonsmokers' Rights Foundation, 2013*; *Australian Associated Press, 2013*; *Non-Smokers' Rights Association, Canada*). This is also so in New Zealand, where in addition the Government has a goal of a smokefree nation by 2025 (*New Zealand Government, 2011*). In the light of this background, we aimed to pilot the use of observational methods for the collection of data on smoking behavior and butt disposal at bus stops.

## METHODS

The study area covered Wellington and Lower Hutt cities with a mix of convenience sampling (as per the researchers' routine travel to work), and purposeful sampling of bus stops in the central business districts that were along major urban bus routes. Two observers separately collected data at different sites on 10 days between 16 September to 11 October 2013 (spring in New Zealand). This was at a total of 11 bus stops at various times in the time period from 06:00 h to 19:00 h. Observations were only done at times when it was not raining.

The observational methods used in a previous observational study of cigarette butt disposal (*Patel, Thomson & Wilson, 2013*), were adapted for the bus stop setting. People seen smoking were observed systematically with the following data items recorded: gender, estimated age-group (under 20, 20–49, 50+ years), type of cigarette (factory made/roll-your-own), position in relation to any bus shelter (inside/outside), means of any cigarette butt extinguishing, and the means of butt disposal (bin, "re-packet", footpath, in bus shelter, road, gutter, vegetation). The presence of rubbish bins was also recorded (less/greater than five m from the smoker, or no bin within sight of the smoker). To measure potential SHS exposure to other people, the presence and number of any young people (<20 years) and adults was recorded when smoking was first observed, and the distance between smokers and others was estimated. Data were entered onto a preformatted paper form or as text into a mobile phone (the former is available on request from the authors).

Pilot testing of the revised list of data collection items was conducted in late 2012 ($n = 18$ smoking events). We decided against performing a formal assessment of inter-rater reliability since this was found to be high in the previous observational study of cigarette butt disposal we had conducted (*Patel, Thomson & Wilson, 2013*).

The definition of a "bus stop area" was the pavement or shelter space within 10 m of the dashed yellow lines on the road denoting the area at which a bus could stop. People in this area who were smoking were only systematically observed if they were standing or sitting (i.e., apparently waiting for a bus or purposefully using the area as a place to smoke). That is, we ignored people smoking who walked through the bus stop area on the way to somewhere else.

Data were compiled in Microsoft Excel and analyzed in R, version 2.13.2 © 2011. Ethical approval for this research was obtained through the ethics approval process of the University of Otago (with no requirement for consent in this observational study).

## RESULTS

This observational method appeared to be feasible with no evidence that the people at the bus stops were aware of being observed and there were no concerns around observer safety for either the female or male observers.

During 27 h of observation across 11 bus stops, we observed a total of 112 cigarettes being smoked by 104 smokers (Table 1). Smoking occurred in the presence of: just adults (46%), both young people and adults (44%), just young people (6%) and the smoker alone (5%) (Table 1). In bus stops that included an enclosed shelter, 33% of the cigarettes smoked occurred with the smoker in the shelter with others present, rather than smoking outside of the shelter.

When a person was observed smoking, there was an average of 6.3 adults and 3.8 young people present at the bus stop (additional tabulated data available on request). When smoking occurred in enclosed bus shelters ($n = 14$), an average of 9.2 young people and 8.5 adults were also inside the shelter (with 129 young people and 119 adults being exposed to this smoking, including some repeat counts of the same people who were exposed to multiple smoking events).

The estimated mean minimum distance between two people smoking simultaneously was fairly similar to the observed minimum between smokers and adults who didn't smoke (1.3 m versus 1.7 m, $p = 0.130$, two sample t-test). Smokers kept a non-significantly greater minimum distance from young people relative to adults who were not smoking (2.2 m versus 1.7 m, $p = 0.214$, two sample t-test). But when at least one non-smoking adult and at least one young person were present, the smoker was, on average, significantly closer in terms of minimum distance (by 0.8 m) to the adult than they were to the young person ($p = 0.038$, paired t-test).

Littering was the major form of cigarette disposal at 84% of cigarettes smoked (95% CI; 77%–90%), the remaining being put in a bin (13%) or returned to the packet ("re-packeted") (3%). The latter were cigarettes extinguished on bus arrival, presumably for subsequent re-use.

**Table 1** Observed smoking and cigarette disposal at bus stops (Wellington and Hutt Cities).

| Aspect of smoking behavior | All cigarettes observed being smoked | | |
|---|---|---|---|
| | n | % | 95% CI for the % |
| **Presence of others at all bus stops** (n = 11 stops) | | | |
| Smoking with both young people and adults present | 49 | 43.8 | 34.9–53.0 |
| Smoking with just young people present | 7 | 6.3 | 3.1–12.3 |
| Smoking with just adults present | 51 | 45.5 | 36.6–54.8 |
| Smoking alone[a] | 5 | 4.5 | 1.9–10.0 |
| **Total** | **112** | **100** | |
| **Bus stops with an "enclosed shelter"** [b] (n = 5 stops) | | | |
| Smoking in shelter with others present | 14 | 33.3 | 21.0–48.5 |
| Smoking in a shelter alone[a] | 0 | 0.0 | 0.0–8.4 |
| Smoking outside the shelter with others present | 24 | 57.1 | 42.2–70.9 |
| Smoking outside the shelter alone | 4 | 9.5 | 3.8–22.1 |
| **Sub-total** | **42** | **100** | |
| **Bus stops with "partial shelter"** [b] (n = 3 stops) | | | |
| Smoking with others present | 51 | 98.1 | 89.9–99.7 |
| Smoking alone[a] | 1 | 1.9 | 0.3–10.1 |
| **Sub-total** | **52** | **100.0** | |
| **Outdoor bus stops** (n = 3 stops) | | | |
| Smoking with others present | 18 | 100 | 82.4–100.0 |
| Smoking alone[a] | 0 | 0.0 | 0.0–17.6 |
| **Sub-total** | **18** | **100** | |
| **Disposal site for the cigarettes** | | | |
| Not littered – Rubbish bin | 15 | 13.4 | 8.3–21.0 |
| Not littered – "Re-packeted" (extinguished at bus arrival) | 3 | 2.7 | 0.9–7.6 |
| Littered – Footpath | 62 | 55.4 | 46.1–64.2 |
| Littered – In bus shelter (e.g., on the ground) | 12 | 10.7 | 6.2–17.8 |
| Littered – Road | 9 | 8.0 | 4.3–14.6 |
| Littered – Gutter/drain | 6 | 5.4 | 2.5–11.2 |
| Littered – In vegetation | 5 | 4.5 | 1.9–10.0 |
| **Total** | **112** | **100** | |
| **Littering and rubbish bin access** | | | |
| Not littered (used bin or "re-packeted") | 18 | 16.1 | 10.4–24.0 |
| Littered with bin within 5 m | 52 | 46.4 | 37.5– 55.6 |
| Littered with bin >5 m but bin was potentially within the visual field of the smoker (assuming normal vision) | 42 | 37.5 | 29.1–46.7 |
| Littered with no bin in the potential visual field of the smoker | 0 | 0.0 | 0.0–3.3 |
| **Total** | **112** | **100** | |

[a] The observer was sometimes technically inside the bus stop area, but also at times just outside of this area. Nevertheless, the observer was not included in any of the data in this table.

[b] An "enclosed" bus shelter was defined as having three sides and a roof (and usually the side open to the road was partially walled). These shelters are typically around 2 m by 3 m in area. Bus stops with "partial shelter" were defined as those with a roof and only one side (i.e., usually an overhang from a shop or other building).

Most littered cigarette butts were not extinguished (65%; 61/94), and 4% were discarded into vegetation (Table 1). Littering appeared more common with lower smoker age over the three age groups considered (but not at a significant level; Chi square for trend: $p = 0.10$). All littering occurred with either a rubbish bin close by (i.e., 55% or

52/94 within 5 m) or further away but still visible (45% or 42/94). There were no statistically significant associations with littering relating to: smoker gender, type of cigarette (manufactured or roll-your-own), time of day, weekday versus weekend, central city versus suburb, or Wellington versus Hutt cities.

## DISCUSSION

This pilot study is limited by its small size and the type of sampling approaches used (both convenience sampling and purposeful sampling of busy bus stops). These methods were adopted to maximize the efficiency of data collection in what was a time-limited and unfunded study. Other limitations with these observational data include estimating the age-groups of observed people (albeit in broad categories), and estimating the minimal distances between the smoker and others. In some cases it was suspected that the same individual smoker was counted at a subsequent observation session at the same bus stop (but this could not be precisely documented as no photographs were taken). Future work could include inter-rater reliability around such aspects as age-group and distance assessment.

Despite these limitations, this study appears to be the first published attempt (to our knowledge), of providing this extent of observational evidence around smoking at bus stops. We found that smokers usually (95% of the time), smoked around others and that it occurred relatively closely to both young people and adults. It also sometimes occurred within highly enclosed bus shelters where the SHS was likely to accumulate. Also over half the smoking was with young people present, which probably has implications for making smoking appear normal to those who are at risk of becoming future smokers, i.e., given some evidence for smoking denormalization for youth with smokefree areas in data from the United Kingdom (*Rooke et al., 2013*). Indeed, there is evidence that smokefree areas partly work through smoking denormalization, along with reducing opportunities to smoke (*IARC, 2009*).

There was relatively little evidence that people smoking were purposefully limiting their proximity to other people (just the significantly greater minimum distance from young people). The proximity between people smoking and others is problematic from a health and nuisance perspective, especially given data around the nuisance issue for public transport users (*Russell, Wilson & Thomson, 2012*), and around high levels of particulates when smoking within partly enclosed bus shelters (*Patel, Thomson & Wilson, 2012*). Some of these people at a bus stop with a person smoking may have minimal exposure to SHS if the wind direction is favorable for them, but even people "upwind" can be exposed. Recent research on fine particulates (PM2.5) indicates that elevated levels of SHS can travel at least 9m both downwind and upwind from a single burning cigarette (when average wind speed is 0.8 m/s, albeit with higher levels [2.5x] for downwind) (*Hwang & Lee, 2013*).

The majority of smokers at bus stops littered their cigarettes (84%), similar to the finding of a previous observational study in Wellington City at 77% (*Patel, Thomson & Wilson, 2013*), and a US study at 56% (albeit for self-reporting) (*Rath et al., 2012*). These

differences with the US study may suggest that self-reported data potentially underestimates littering levels (and indeed this might be expected due to social desirability bias and because such littering is often illegal). Nevertheless, the occurrence of littering is relevant in terms of increased cleaning costs, fire risk and pollution of waterways (where storm-water drains carry butts into these waterways). Also the presence of rubbish bins at bus stops seemed to be having a minimal effect on influencing smoker behavior in our study. One explanation from research elsewhere is that most smokers do not conceptualize butts as litter (*Rath et al., 2012*).

While more research on these issues with larger studies is desirable, policy makers in many jurisdictions could follow the lead of some US, Canadian and Australian cities and states to act now to reduce the problems associated with smoking at bus stops. Such action can be justified in terms of reducing exposure to SHS, reducing nuisance effects and reducing littering. Improving the quality of the public transport experience for the majority of users (who are non-smokers) could also be part of the considerations from a wider health perspective. Possible policy responses include: (i) smoker education around not smoking near others and not littering; (ii) increasing fines and enforcement around littering; and (iii) instituting national laws or local by-laws for requiring smokefree transportation settings. Of these, the most effective and cost-effective for most jurisdictions will probably be new smokefree laws covering transportation settings. Indeed, for the New Zealand setting at least, there is survey evidence to suggest that there is majority public support for the expansion of smokefree areas. That is, 76% (54% of smokers) agreed that "smoking should be banned in all outdoor public places where children are likely to go" (*Trappitt, Li & Tu, 2011*). There are however, no New Zealand survey data specifically on public attitudes to smokefree transportation settings.

## ACKNOWLEDGEMENT
We thank Dr Nevil Pierse for providing statistical advice.

### Funding
This study had no external funding.

### Author Contributions
- Nick Wilson conceived and designed the experiments, performed the experiments, analyzed the data, contributed reagents/materials/analysis tools, wrote the paper, prepared figures and/or tables, reviewed drafts of the paper.
- Jane Oliver performed the experiments, analyzed the data, contributed reagents/materials/analysis tools, wrote the paper, prepared figures and/or tables, reviewed drafts of the paper.
- George Thomson conceived and designed the experiments, contributed reagents/materials/analysis tools, wrote the paper, reviewed drafts of the paper.

**Peer**J

## Human ethics

The following information was supplied relating to ethical approvals (i.e., approving body and any reference numbers):

University of Otago, approved under Category B Ethical Approval on 13/9/12.

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
