# Peer review of "Smoking close to others and butt littering at bus stops: pilot observational study"

_PeerJ, doi:10.7717/peerj.272_

## Round 0.1 · original submission · Minor Revisions

Dear Authors - please find attached your peer review.

·

Basic reporting

Introduction: While the introduction provides sufficient background, there are some awkward sentences and flow issues. I have made some recommendations below.

Authors report that SHS exposure at bus stops or train stations is “relatively common” in NZ, based on 11.9% of respondents reporting such exposure in last month from national survey data. I would suggest removing “relatively common”, which is quite subjective and it could be argued that 12% is “relatively uncommon”. Instead could word as: The NZ national survey found that 11.9% of respondents reported exposure to SHS at a bus stop or train station in past month. OR change “relatively common” to “significant”, although one could argue what is to be considered “significant”.

In par 3, awkward sentence, suggest rewording “Another problem is smoking-related litter at bus stops which contributes to cleaning costs, fire risks and may make the public transport using experience less attractive.”
To: Another problem is smoking-related litter at bus stops which contributes to cleaning costs, fire risks and may make the [experience of using public transport less attractive].

Par 4 is awkward, suggest introducing the following 2 sentences about the goals of the NZ to be smokefree in first paragraph with some rewording as follows:
“Transportation settings such as bus stops and train station platforms are increasingly being covered by smokefree legislation in various jurisdictions...
However, there are currently no smokefree laws or by-laws that restrict smoking at bus stops in any New Zealand cities. This is despite the goal of the New Zealand government to be a smokefree nation by 2025...”

In final paragraph, suggest adding a sentence after the aim on the relevance of the study to addressing exposure, e.g., “Given this background, we aimed to pilot the collection of data on smoking behavior and butt disposal at bus stops.” [The results will inform...].

Experimental design

The term “relatively busy” is problematic- how was this determined? Can a more specific definition of ‘business’ be used, i.e., main routes or hubs, number of buses, or observed to have x number of riders, etc.

The description of observation sessions/times is too vague. What do you mean by “episodic”, i.e., how many episodes were conducted at each site and were the sessions conducted in the same day or on different days? How long was each observation episode? Were sites only visited in good weather or in all weather situations, as rain or cold might impact behaviour (i.e., in rain there could be more smokers using the bus shelters).

Did the 2 observers separately collect data at same site at same time, or did they split up the sites? Assuming the former, but needs to be clarified.

Awkward sentence: “We decided against performing a formal assessment of inter-rater reliability as [the score was high in the previous observational study of butt disposal that] we had conducted..

Validity of the findings

“During 27 hours of observation [across all sites], we observed [a total of] 112 cigarettes being smoked by 104 smokers...”

Additional comments

An important consequence of visual exposure to smoking is the normalization of smoking, particularly among susceptible young people. In the discussion, the authors state: “Over half the smoking was with young people present, which also has implications for making smoking appear normal to those who are at risk of becoming future smokers.” However, at least one reference should be cited and more emphasis placed on this in the discussion.

If available, national data on opinions about smoking bans at bus stops would help to provide context in the background section and could also be introduced in the discussion.

Otherwise, I believe the research to be relevant and meaningful to smoke-free policy development!

·

Basic reporting

No comments

Experimental design

The experimental design uses convenience and purposive sampling which is appropriate for a pilot study of this nature. Moreover, the observation methods were already piloted (Patel et al., 2013) which strengthen their validity for use in this study.

Validity of the findings

The only concern is determining estimated age-group (under 20, 20-49, 50+ years) which is inherently biased based on the observer and begs the accuracy of the measure (this is especially true if there was not inter-rater indices on this estimate).

This also has implications for the validity of the findings, given that there is a distinction made between age groupings. This issue may be clarified if the authors clearly explain how they defined young people and adults.

However, the authors have noted this potential inaccuracy in the estimation of age as a limitation in the discussion section. I suggest that the authors include a sentence that such a limitation could be minimized/mitigated by inter-rater tests in future studies--the same issue with the estimation of minimal distances between smokers and others.

Additional comments

None

·

Basic reporting

This is a very small observational study...more of a pilot than a robust research activity. The authors acknowledge this, and also the limitations of the observational activities, sampling procedures, etc. Still, it is interesting and somewhat novel in terms of identifying needs for future studies and policy applications.

Experimental design

This is an observational study, with only some very basic descriptive data reported. There was no hypothesis to test, and the research question was not clearly stated. The authors called upon a previous observational study on cigarette butt disposal (Patel 2013) and thus are to be commended for following through on this particular line of investigation. It is not clear how rigorously the observers followed protocols, and perhaps the actually data collection guides could be included as an on-line annex. This would allow reproduction of results. One criticism is that inter-observer reliability was not tested.

Validity of the findings

Given the limitations of the sample and observational methods, the results reported have value as a pilot study. Appropriate statistical measures were incorporated into the analyses, and these added value to what might be considered observational exposure assessments. The descriptive data on non-use of butt waste receptacles are quite interesting, and demonstrate that self-reported littering behavior probably underestimates the real-time behavior of most smokers. This might be brought out more specifically in the discussion, in fact.

Additional comments

This is a nicely written paper, following on previous observational studies of butt littering,and demonstrating the need for policy implementation at bus stops and other transportation venues to prevent SHS exposure. The authors may bring out the non-compliance with waste bin use for discarded butts a bit more in the discussion. This shows that self-reported littering by smokers is probably underestimated in various reports.

---

## Round 0.2 · Minor Revisions

Please attend to the minor revisions highlighted by the reviewers

·

Basic reporting

Line 59...'and' is repeated twice.

Experimental design

No further comments

Validity of the findings

No further comments

Additional comments

No further comments

·

Basic reporting

no comments

Experimental design

no comments

Validity of the findings

no comments

Additional comments

I am pleased that the authors provided some further information in the discussion regarding smoking denormalization. However, reference(s) should be included to support the statement: "...there is some evidence for smoking denormalization for youth with smokefree areas in data from the United Kingdom", and "... smokefree areas partly work through smoking denormalization, along with reducing opportunities to smoke".

Also, glad to see the addition of public support data, which appears to come from the National Survey, but should be clarified and referenced.

---

## Round 0.3 · accepted · Accept

Congratulation - happy to publish your paper - thanks for your swift responses to the reviewers comments.